# UNSEEN ACTION RECOGNITION WITH UNPAIRED ADVERSARIAL MULTIMODAL LEARNING

## ABSTRACT

In this paper, we present a method to learn a joint multimodal representation space that allows for the recognition of unseen activities in videos. We compare the effect of placing various constraints on the embedding space using paired text and video data. Additionally, we propose a method to improve the joint embedding space using an adversarial formulation with unpaired text and video data. In addition to testing on publicly available datasets, we introduce a new, large-scale text/video dataset. We experimentally confirm that learning such shared embedding space benefits three difficult tasks (i) zero-shot activity classification, (ii) unsupervised activity discovery, and (iii) unseen activity captioning.

## 1 INTRODUCTION

Videos contain multiple data sources, such as visual, audio and text/caption data. Each data modality has distinct statistical properties capturing different aspects of the event. Current state-of-the-art activity recognition models (Carreira & Zisserman, 2017; Tran et al., 2017) only take visual data and class labels as input, which limits the information the model can learn from. For example, the sentence 'a group of men play basketball outdoors' contains rich information, such as 'outdoors' and 'group of men' compared to just the activity class of 'basketball.' Without multimodal learning, these models are unable to benefit from the additional information.

In this paper, we explore multimodal learning from video and language data, each having its own representation. Video data is represented as a sequence of images (spatio-temporal pixel data) while text is represented as a sequence of word embeddings (temporal data). Using a shared embedding space allows for learning the highly non-linear relationships between these modalities. The goal of learning such relationships is to capture similarities between concepts (e.g., basketball and volleyball both being sports with a ball) and further generalize to concepts not seen during training. To enable this generalization, we rely on pre-trained word embeddings as they capture relationships between all words in the English language (Mikolov et al., 2013). By learning a joint representation space, we can capture these relationships, allowing generalization to unseen video activities.

To evaluate our approach, we test on several tasks based on recognizing activities not seen during training. First, we test on a zero-shot learning task, where we classify activity classes not seen during training. Our second task is unsupervised activity discovery, where no class information is provided during training, relying only on the relationship between text and videos to learn good embeddings. Finally, we evaluate on captioning videos with activities/words not seen during training.

Many existing approaches to both zero-shot and embedding space learning require paired data examples, which can be expensive to obtain. By taking advantage of adversarial learning (Goodfellow et al., 2014), we are able augment our method with unpaired data (i.e., random sentences and random videos without any labels or correspondence) to further improve our learned shared embedding space.

In this paper, we design a method capable of learning joint video/language embedding spaces using both paired and unpaired data and experimentally confirm its benefit to three challenging tasks (i) zero-shot activity recognition, (ii) unsupervised activity discovery, and (iii) unseen activity captioning. We show that the use of unpaired, multimodal data allows learning a shared embedding space that generalizes to unseen data better than existing approaches.

## 2 RELATED WORKS

**Multimodal learning** Previous approaches to multimodal learning have used Restricted Boltzmann Machines (Srivastava & Salakhutdinov, 2012) or log-bilinear models (Kiros et al., 2014) to learn distributions over sentences and images. Ngiam et al. (2011) designed an autoencoder that learns joint audio-video representations, however they are unable to train the model end-to-end, instead relying on greedy, layer-by-layer training and further do not evaluate the effect of using unpaired data. Similarly, Chandar et al. (2016) proposed a auto-encoder able to learn correlations between different view of images. Frome et al. (2013) describe a model that maps images and words to a shared embedding. However, these works either learn a joint embedding by concatenating the different features or require a triplet consisting of positive and negative matches and do not evaluate the effects of using unpaired data.

**Text and vision** Using both text and visual data has been studied for many tasks, such as image captioning (Karpathy et al., 2014; Johnson et al., 2016; Karpathy & Fei-Fei, 2015) or video captioning (Krishna et al., 2017; Zhou et al., 2018; Xu et al., 2018). Other works have explored the use of textual grounding for image/video retrieval (Gupta & Mooney, 2010; Rohrbach et al., 2016; Miech et al., 2017; Hendricks et al., 2017). There have been various models proposed to learn a fixed text embedding space with mappings from video features into this embedding space (Guadarrama et al., 2013; Otani et al., 2016; Song et al., 2016; Wang et al., 2016; Xian et al., 2016). However, these works all learn only a single directional mapping (e.g., only mapping from visual to text), only learn with paired text/image samples and require additional data in the form of positive/negative pairs.

**Learning with unpaired data** Recently, there have been many works taking advantage of variational autoencoders (VAEs) (Kingma & Welling, 2014) or generative adversarial networks (GANs) (Goodfellow et al., 2014) to learn mappings between unpaired samples. CycleGan (Zhu et al., 2017) uses a cycle-consistency loss (i.e., the ability to go from a sample in one domain to a second domain then back to the source) to learn unpaired image translation (e.g., image to sketch). Other works learn many-to-many mappings between images (Almahairi et al., 2018) or use two GANs to map between domains (Yi et al., 2017). An autoencoder with shared weights for both domains has been used to learn a latent space for image-to-image translation (Liu et al., 2017). However, these works all focus on learning mappings between unpaired data of the same modalitiy (e.g. image to image), where the data is from the same underlying distribution. Our work focuses on learning from different modalities with very different distributions.

**Zero-shot activity recognition** Many previous works have studied zero-shot activity recognition. Common approaches include using attributes (Liu et al., 2011; Palatucci et al., 2009; Romera-Paredes & Torr, 2015) or word embeddings (Xu et al., 2015; 2017; Norouzi et al., 2013; Socher et al., 2013; Kodirov et al., 2017) or learning a similarity metric (Zhang & Saligrama, 2015; Chopra et al., 2005). Some works have explored using adversarial losses on the latent space (Chen et al., 2018) or use GANs to generate features for unseen classes (Xian et al., 2018) or used auto-encoders (Wang et al., 2017).

Our work differs from these previous works in three key ways: (1) we show the benefit of additional data augmentation using unpaired samples, (2) we experimentally compare the use of the embedding space for both zero-shot recognition, unseen recognition, and unseen video captioning, and (3) we learn a shared, multimodal embedding space with bi-directional mappings in an end-to-end fashion.

## 3 METHOD

To enable learning of a joint embedding space, we use a deep autoencoder architecture. Our model consists of 4 neural networks:

$$\textbf{Video Encoder } E_V : v \mapsto z_v \qquad \textbf{Video Decoder } G_V : z \mapsto v$$
$$\textbf{Text Encoder } E_T : t \mapsto z_t \qquad \textbf{Text Decoder } G_T : z \mapsto t$$

where $v$ is a sequence of video data and $t$ is a sentence (sequence of words). $z$ is the embedded data in the shared space that we are learning. The encoders learn a compressed representation of the video or text while the decoders are trained to reconstruct the input:

$$\mathcal{L}_{recons}(v,t) = ||G_V(E_V(v)) - v||_2 + ||G_T(E_T(t)) - t||_2 \qquad (1)$$

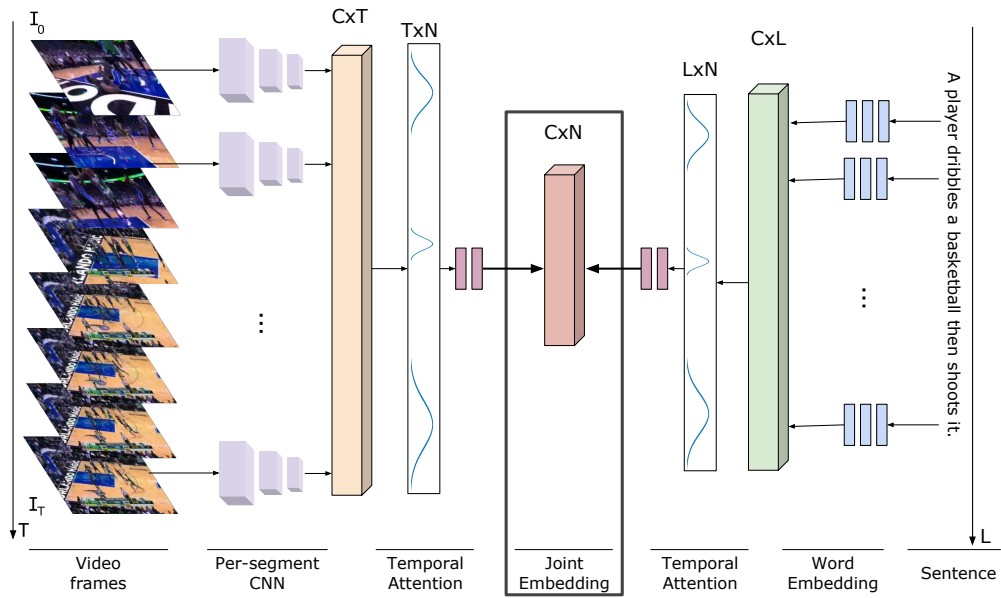

Figure 1: Illustration of the encoder models used to learn a joint embedding space. Videos and sentences are mapped into a low-dimensional space by applying CNNs and temporal attention. Then several fully-connected layers map into the joint embedding space. The decoders follow this same architecture with the weights transposed.

As both text and video data are sequences, they often have different lengths. A joint embedding space requires that the features from both modalities have the same dimensions. Given a text representation of length $L$ and a video representation of length $T$, we need to obtain a fixed-size representation. To learn a fixed-dimensional embedding, there are many choices for the encoder/decoder architecture, such as temporal pooling (Ng et al., 2015), attention (Piergiovanni et al., 2017) or RNNs (Krishna et al., 2017). We chose temporal attention filters (Piergiovanni et al., 2017) as they learn a mapping from any length input to a $N$-dimensional vector and have been shown to outperform temporal pooling and RNNs on activity recognition tasks.

The attention filters consists of $N$ Gaussians, each learning 2 parameters: a center $\hat{g}$ and width $\sigma$. The filters are determined by:

$$
\begin{aligned}
g_n &= 0.5 \cdot T \cdot (\widetilde{g}_n + 1) \\
\mu_n^i &= g_n + (i - 0.5N + 0.5) \\
F[n, t] &= \frac{1}{Z} \exp(-\frac{(t - \mu_n)^2}{2\sigma_n^2}) \\
n &\in \{0, 1, \ldots, N-1\},\ t \in \{0, 1, \ldots, T-1\}
\end{aligned}
\tag{2}
$$

The weights are applied by matrix multiplication with the video or text sequence: $v' = Fv$. Additionally, we can learn a transposed version of these filters to reconstruct the input: $v = F^T v'$. To reconstruct the input, the decoders learn their own parameters with the tensors transposed, resulting in the matching output size. The architecture of the encoders is shown in Fig. 1.

### 3.1 LEARNING A JOINT EMBEDDING SPACE

To learn a joint representation space, we minimize the $L_2$ distance between the embeddings of a pair of text and video (shown in Fig. 2(a)):

$$
\mathcal{L}_{joint}(v, t) = ||E_V(v) - E_T(t)||_2
\tag{3}
$$

This forces the joint embeddings to be similar and when combined with the reconstruction loss, ensures that the representations can still reconstruct the input.

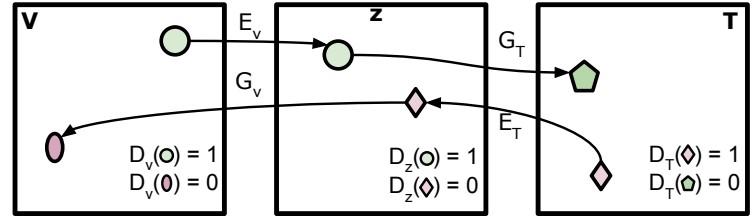

Figure 2: Visualization of several constrains on the shared embedding space. Circles are video data, ovals are reconstructed video. Diamonds are text data, and pentagons are reconstructed text. **(a)** The reconstruction (Eq. 1) and joint (Eq. 3) losses. **(b)** Mapping from text to video using the cross-domain (Eq. 4) loss.

Figure 3: Visualization of the adversarial formulation to learn with unpaired data. We create 3 discriminators, (1) $D_z$ learns to discriminate examples of text/video in the latent space. (2) $D_V$ learns to discriminate video generated from text compared to video. (3) $D_T$ learns to discriminate generated text compared to text.

We can further constrain the networks and learned representation by forcing a cross-domain mapping from text to video and from video to text (shown in Fig. 2(b)):

$$\mathcal{L}_{cross}(v,t) = ||G_T(E_V(v)) - t||_2 + ||G_V(E_T(t)) - v||_2 \qquad (4)$$

Additionally, we can use a 'cycle' loss to map from video to text and back to video. Note that while the previous losses all require paired examples, this loss does not.

$$\mathcal{L}_{cycle}(v,t) = ||G_T(E_V(G_V(E_T(t)))) - t||_2 + ||G_V(E_T(G_T(E_V(v)))) - v||_2 \qquad (5)$$

To train the model to learn a joint embedding space, we minimize

$$\mathcal{L}(v,t) = \mathcal{L}_{recons}(v,t) + \alpha_1\mathcal{L}_{joint}(v,t) + \alpha_2\mathcal{L}_{cross}(v,t) + \alpha_3\mathcal{L}_{cycle}(v,t) \qquad (6)$$

where $\alpha_i$ are hyper-parameters weighting the various loss components.

### 3.2 Semi-supervised learning with unpaired data

To learn using unpaired data (i.e., unrelated text and video), we use an adversarial formulation. We treat the encoders and decoders as generator networks. We then learn an additional 3 discriminator networks which constrain the generators and embedding space and force the encoders and decoders to be consistent:

(1) $D_z$ which learns to discriminate between latent text representations and latent video representations. Conceptually, this constrains the learned embeddings to appear to be from the same distribution.

(2) $D_V$ which learns to discriminate between true video data and generated video data $G_V(E_T(t))$.

(3) $D_T$ which learns to discriminate between true text data and generated text data, $G_T(E_V(v))$.

Given these discriminators, we minimize the following losses:

$$\begin{aligned}
\mathcal{L}_{D_z}(v,t) &= -\log(D_z(E_T(t))) - \log(1 - D_z(E_V(v))) \\
\mathcal{L}_{D_V}(v,t) &= -\log(D_V(v) - \log(1 - D_V(G_V(E_T(t)))) \\
\mathcal{L}_{D_T}(v,t) &= -\log(D_T(t)) - \log(1 - D_T(G_T(E_V(v))))
\end{aligned} \qquad (7)$$

Using the discriminators, we can train the generators (encoders and decoders) to minimize the following loss based on unpaired data:

$$\mathcal{L}_{G_z}(v,t) = \log(D_z(E_T(t))) + \log(1 - D_z(E_V(v)))$$
$$\mathcal{L}_{G_V}(v,t) = \log(1 - D_V(G_V(E_T(t)))) \tag{8}$$
$$\mathcal{L}_{G_T}(v,t) = \log(1 - D_T(G_T(E_V(v))))$$

Note that in this formulation, $v$ and $t$ are not paired.

These networks are trained in an adversarial setting. For example, for the text-to-video generator (i.e., $v' = G_V(E_T(t))$ and video discriminator, $D_V$, we optimize the following minimax equation:

$$\min_{E_T, G_V} \max_{D_V} = \mathbb{E}_{v \sim p_{\text{data}}(v)}[\log D_V(v)] + \mathbb{E}_{t \sim p_{\text{data}}(t)}[\log(1 - D_V(G_V(E_T(t))))] \tag{9}$$

This equation is similarly applied for video-to-text. For learning the embedding space with the video and text encoders, $E_V, E_T$ and the discriminator $D_z$, we optimize the following minimax equation:

$$\min_{E_T, E_V} \max_{D_z} = \mathbb{E}_{v \sim p_{\text{data}}(v)}[\log D_z(E_V(v))] + \mathbb{E}_{t \sim p_{\text{data}}(t)}[\log(1 - D_z(E_T(t)))] \tag{10}$$

As training GANs can be unstable, we developed an method to allow for more stable training of the joint embedding space, shown in Algorithm 1. We initialize both the generator and discriminator networks by training only on paired data. After several iterations of this, we train with both unpaired and paired data. We found the initial training of the generators and discriminators was important for stability, without it the loss often diverges and the learned embedding did not generalize to unseen activities.

---

**Algorithm 1** Semi-supervised alignment with adversarial learning

---

**function** TRAIN
    **for** number of initialization iterations **do**
        Sample $(V, T)$ from pair training data
        Update encoders/decoders based on paired data (Eq. 6)
        Update discriminators (Eq. 7)
    **end for**
    **for** number of training iterations **do**
        Sample $(V_p, T_p)$ from paired and $(V_u, T_u)$ from unpaired training data
        Update encoders/decoders based on paired data (Eq. 6)
        Update encoders/decoders based on unpaired data (Eq. 8)
        Update discriminators based on all samples (Eq. 7)
    **end for**
**end function**

---

## 4 EXPERIMENTS

We compare our various approaches on different tasks (i) zero-shot activity recognition, (ii) unsupervised activity discovery and (iii) unseen activity captioning. These tasks test various combinations of our encoders and decoders and how well the shared representation generalizes to unseen data. We experimentally confirm the benefits of our methods using multiple public datasets: AcitivtyNet (Heilbron et al., 2015; Krishna et al., 2017), HMDB (Kuehne et al., 2011), UCF101 (Soomro et al., 2012), and MLB-YouTube (Piergiovanni & Ryoo, 2018).

### 4.1 IMPLEMENTATION/TRAINING DETAILS

We implement our models in PyTorch. For the per-segment video CNN, we use I3D (Carreira & Zisserman, 2017) to obtain a $1024 \times T$ video representation. We trained a version of I3D based on Kinetics-600, but withheld all classes that appear in ActivityNet, HMDB51, or UCF101 so that the classes are truly unseen. This resulted in a training set with 478 classes and 278k videos. Since generating videos is an extremely challenging task, the video autoencoders start with and generate

the I3D feature. We use GloVe word embeddings (Pennington et al., 2014) to obtain a language representation. We set $N = 4$ for the temporal attention filters and apply 4 fully connected layers. These layers are followed by $L_2$ normalization so that the embedding space has unit length (Tygert et al., 2015). We train for 200 epochs and use stochastic gradient descent with momentum to minimize the loss function with a learning rate of 0.01. After every 50 epochs, we decay the learning rate by a factor of 10. When training in the adversarial setting (e.g., Algorithm 1), we initialize the network training for 50 epochs on paired data followed by 200 on the paired + unpaired data.

**Baselines**   For baselines, we compare to a fixed-text embedding space, were only a mapping from video data into the text embedding space is learned (e.g., (Otani et al., 2016)). We also compare to learning a shared embedding space with the 'recons' (Eq. 1) and 'cross' (Eq. 4) terms (e.g., (Ngiam et al., 2011)).

## 4.2   Zero-shot activity recognition

Zero-shot activity recognition is the problem of classifying a video that belongs to a class not seen during training. Given training videos of seen classes together with paired text descriptions, our approach learns a shared embedding that maps videos/texts from multiple seen classes. The objective is to classify videos of unseen classes solely based on the learned embedding space and the text samples.

We use the ActivityNet captions (Krishna et al., 2017) dataset to learn the joint embedding space, as this dataset has both sentence descriptions for each video as well as activity classes. We randomly choose a set of $K$ activity classes and withhold all videos/sentences belonging to those classes during training. For testing, we take a subset of sentences for the unseen classes and map the sentences into the joint embedding space, $z_t = E_T(t)$. We then map the videos into the space, $z_v = E_V(v)$ and use nearest neighbors to match each video ($z_v$) to text ($z_t$), using the class of the nearest sentence as the classification for the video.

In Table 1, we compare the effect of the various loss components. For each method, we run 10 trials each with a different set of unseen activity classes and average the results. We find that previous methods of learning a fixed language embedding (e.g., (Socher et al., 2013; Xu et al., 2015; 2017)) are significantly outperformed by learning a joint representation. Previous methods learning embedding spaces without the 'joint' term (e.g. (Ngiam et al., 2011)), we found yield nearly random performance on these tasks, suggesting that forcing the representations to match in the embedding space is important. Further, adding the reconstruction, cross-domain, and cycle losses all improve performance. We also compare to a standard triplet loss (e.g., (Guadarrama et al., 2013)) which requires additional positive/negative samples. We find that the triplet loss outperforms the 'joint' loss, but is surpassed by adding the 'cycle' and 'cross' terms, which use less data.

We also compare the various components of the adversarial loss. We compare to having just the adversarial loss on the embedding space ($D_z$), like (Chen et al., 2018), and compare just the adversary on the generated videos/sentences. We find that the use of all terms in important for performance.

To obtain unpaired data, we use the sentence descriptions from the Charades (Sigurdsson et al., 2016) dataset, which contains many activities in a home setting. The unpaired video data is sampled from HMDB and UCF101. While previous works such as (Ngiam et al., 2011) can support learning with unpaired data, we find that the adversarial loss provides better results than just the 'cycle' and 'recons' terms, and further improves over training with just paired data.

In Table 2, we compare our approach to previous zero-shot learning methods on HMDB and UCF101. The paired training data for these models is drawn from ActivityNet with any classes belonging to HMDB or UCF101 withheld. The unpaired text data is sampled from Charades and the video data comes from either HMDB (when testing on UCF101) or UCF101 (when testing on HMDB). As HMDB and UCF101 have no text descriptions, we created a sentence description for each activity class (included in Appendix B). We find that our joint embedding space outperforms the previous approaches on these datasets and unpaired adversarial learning further improves our performance.

Table 1: Comparison of various methods on ActivityNet for 5, 10, 20 or 50 unseen classes. These results are averaged over 10 trials where each trial has a different set of unseen activity classes.

| | 5 Unseen | 10 Unseen | 20 Unseen | 50 Unseen |
|---|---|---|---|---|
| Paired Data | | | | |
| Fixed Text Representation | 41.9 | 38.4 | 29.4 | 15.6 |
| Triplet Loss | 56.8 | 44.9 | 38.8 | 23.3 |
| joint | 54.3 | 41.7 | 36.1 | 21.2 |
| recons + cross | 21.1 | 12.6 | 7.6 | 2.9 |
| joint + recons | 70.1 | 54.4 | 42.6 | 27.5 |
| joint + recons + cycle | 70.4 | 54.3 | 42.1 | 26.8 |
| joint + recons + cross | 72.6 | 55.4 | 43.2 | 27.8 |
| joint + recons + cross + cycle | 76.4 | 56.9 | 45.5 | 28.8 |
| With Adversarial Losses (joint + recons + cross + cycle + Adv.) | | | | |
| $+ D_z$ | 78.4 | 57.3 | 45.9 | 29.1 |
| $+ D_v + D_t$ | 77.2 | 57.2 | 45.6 | 28.9 |
| $+ D_z + D_v + D_t$ | 79.7 | 58.3 | 46.5 | 29.7 |
| Paired + Unpaired Data | | | | |
| recons + cycle | 22.8 | 13.6 | 8.4 | 4.2 |
| joint + recons + cycle | 72.6 | 58.4 | 44.7 | 29.3 |
| joint + recons + cross + cycle | 73.4 | 59.1 | 45.3 | 29.2 |
| All terms | 82.5 | 60.4 | 46.2 | 30.1 |

Table 2: Results on HMDB51 and UCF101 compared to previous state-of-the-art results. We find that learning a shared embedding space is beneficial and that augmented with unpaired data provides the best results.

| | Feature | HMDB51 | UCF101 |
|---|---|---|---|
| SJE (Akata et al., 2015) | IDT | $12.0 \pm 2.6$ | $9.3 \pm 1.7$ |
| ConSe (Norouzi et al., 2013) | IDT | $15.0 \pm 2.7$ | $11.6 \pm 2.1$ |
| ZSECOC (Qin et al., 2017) | IDT | $22.6 \pm 1.2$ | $15.1 \pm 1.7$ |
| Semantic Embedding (Xu et al., 2015) | IDT | $21.2 \pm 3.0$ | $18.6 \pm 2.2$ |
| Manifold Ridge Regression (Xu et al., 2017) | IDT | $24.1 \pm 3.8$ | $22.1 \pm 2.5$ |
| SAE (Kodirov et al., 2017) | I3D | $25.6 \pm 3.2$ | $25.4 \pm 2.2$ |
| Ours (paired) | IDT | $26.3 \pm 3.2$ | $25.4 \pm 3.4$ |
| Ours (paired + unpaired) | IDT | $29.7 \pm 2.2$ | $26.4 \pm 2.1$ |
| Ours (paired) | I3D | $28.3 \pm 2.7$ | $27.8 \pm 2.2$ |
| Ours (paired + unpaired) | I3D | $34.7 \pm 2.4$ | $33.4 \pm 1.8$ |

## 4.3 USE OF UNPAIRED DATA

We explore different strategies for obtaining unpaired data. For these experiments, we keep a fixed set of paired text and videos and explore adding various sources of unpaired data. We compare (i) 10k random sentences from Wikipedia, (ii) 10k random dictionary definitions, and (iii) 10k dictionary

Table 3: Unseen activity recognition results on ActivityNet, HMDB51 and UCF101, evaluated by using both unseen and seen classes for the testing.

| | ActNet (10 unseen) | ActNet (50 unseen) | HMDB51 | UCF101 |
|---|---|---|---|---|
| Fixed Text Representation | 55.7 | 46.8 | 24.5 | 26.8 |
| Triplet Loss | 57.7 | 48.5 | 27.6 | 29.8 |
| joint | 62.1 | 50.2 | 29.8 | 30.6 |
| joint + recons | 64.4 | 52.6 | 30.4 | 31.3 |
| joint + recons + cross + cycle | 69.6 | 58.5 | 35.6 | 36.5 |
| Paired + Unpaired Data | | | | |
| All terms | 71.6 | 65.9 | 38.9 | 42.1 |

Table 4: Comparison of various source of unpaired data on ActivityNet with 10 unseen classes.

|  | Accuracy (unseen) | Accuracy (all classes) |
|---|---|---|
| Paired Data | 58.3 | 69.6 |
| + Random Wikipedia Sentences | 55.8 | 66.4 |
| + Random Dictionary Defs. | 56.3 | 68.2 |
| + Verb Dictionary Defs. | 59.2 | 70.7 |
| + Random YouTube Videos | 58.7 | 70.1 |
| + Verbs + Random Videos | 60.3 | 71.2 |

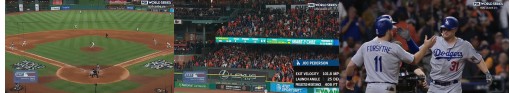

He got right on top of that pitch, Pederson, and shot
and way out of here. Three-run blast.

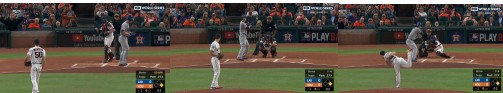

That has been a feat in this series for both teams, nobody is hitting
with two strikes. That's how good the pitching has been.

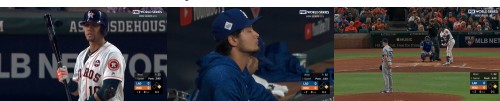

They would suspend him at the beginning of next year
as opposed to for a game during this World Series.

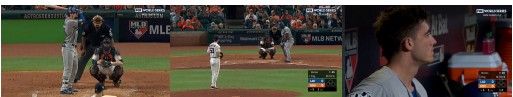

He is an aggressive third baseman and he can really play over
there you know. He definitely takes pride in his defense as well.

Figure 4: Example video sequences from the MLB-YouTube dataset with the commentary caption. **Top**: Sentences that describe the occurring activities. **Bottom**: Sentences that do not describe the current activities.

definitions of verbs. We also compare adding 10k random videos from YouTube as additional video data. We show our results using 10 unseen classes in ActivityNet in Table 4. We find that augmenting with similar unpaired data improves performance, while using irrelevant data harms performance. We find that dictionary verb definitions improve performance the most, as they capture important semantic information regarding the activities we are learning. The use of additional video data is further beneficial.

## 4.4 UNSUPERVISED ACTIVITY DISCOVERY

To further evaluate our joint embedding space, we conducted experiments on unsupervised activity discovery. For this task, we expanded the MLB-YouTube dataset (Piergiovanni & Ryoo, 2018) by densely annotating the videos with a transcription of the announcers' commentary, resulting in approximately 50 hours of aligned text and video. Examples of this data are shown in Fig. 4. The MLB-YouTube dataset is designed for fine-grained activity recognition, where the difference between activities is quite small. Additionally, these captions only roughly describe what is happening in the video, and often contain unrelated stories or commentary on a previous event, making this a challenging task. The dataset will be made publicly available. To train our joint embedding space, we split each baseball video into 30 second intervals and use the corresponding text as paired data, resulting in 6,089 paired training samples.

We evaluate our joint embedding using the segmented videos from MLB-YouTube. For each video, we compute the embedded features and apply $k$-means clustering ($k = 8$, the number of classes). Each segmented video is assigned to a cluster and votes for the cluster label based on its ground truth label. We use that cluster assignment for classification on the MLB-YouTube test set. We report our findings in Table 5. As a baseline, we cluster I3D features pre-trained on Kinetics. We find that our methods improve the embedding space. However, we note that when using unpaired data from Charades, the performance drops. This is likely due to Charades data being very different from MLB-YouTube data. We collected additional captions and baseball videos to augment the MLB-YouTube dataset, and confirmed that unpaired data helps when it is from a similar distribution.

In Table 6 we compare our various methods for unsupervised activity discovery on HMDB and UCF101. Here, we learn a joint embedding space using the ActivityNet videos and captions. We withhold any videos belonging to a class in HMDB or UCF101. Unlike MLB-YouTube, on these datasets, we find that using the unpaired training with Charades further improves performance. This

Table 5: Comparison of unsupervised activity classification on MLB-YouTube.

|  | Accuracy | mAP |
|---|---|---|
| Baseline I3D features | 23.4 | 32.6 |
| Fixed Text Representation | 27.9 | 34.7 |
| Joint | 34.5 | 41.6 |
| Joint + recons | 37.9 | 43.7 |
| Joint + recons + cycle | 44.2 | 48.6 |
| Joint + recons + cross | 43.7 | 49.3 |
| All (paired) | 48.4 | 51.2 |
| All (+ unrelated unpaired) | 39.7 | 43.9 |
| All (+ related unpaired) | 49.1 | 54.3 |

Table 6: Comparison of unsupervised activity classification on HMDB and UCF101.

|  | HMDB | UCF101 |
|---|---|---|
| Baseline I3D features | 26.6 | 42.5 |
| Joint | 32.4 | 57.7 |
| Joint + recons | 33.5 | 59.0 |
| All (paired) | 34.6 | 59.5 |
| All (+ unpaired) | 34.9 | 59.9 |

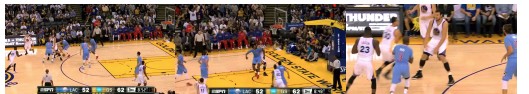

Several men are playing basketball

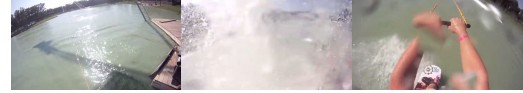

People are swimming in the ocean

Figure 5: Example captions for unseen activities. **Left**: Using a joint embedding space allows the model to correctly caption this video as basketball, despite never seeing an example of basketball during training. **Right**: An example of a caption for the unseen water-ski activity. Here the model fails to correctly caption the activity.

confirms that when the additional data is similar to the target dataset, using the adversarial learning setting further improves the embedding space.

### 4.5 UNSEEN VIDEO CAPTIONING

As our model learns a bi-directional mappings, we can apply our model to generate video captions. Existing video captioning models are unable to create realistic captions for unseen activities, as without training data they do not know the words to describe the video. Given a video, $v$, we can generate a caption by mapping the video to text $t = G_T(E_V(v))$. For each word, we then use nearest neighbors matching with the GloVe embeddings to obtain the words to form a sentence. In Table 7, we report the commonly used METEOR (Banerjee & Lavie, 2005) and CIDEr (Vedantam et al., 2015) scores of our various models, measured with the unseen classes from the ActivityNet dataset. We find that learning a joint representation is beneficial and using unpaired samples further improves the task. In Fig. 5, we show example captioned videos. Note that this task is extremely challenging, as it requires the model to generate captions using activity words (e.g., basketball) not seen during training.

## 5 CONCLUSION

We proposed an approach to learn a joint language/text embedding space using various constraints. We further extended the model to be able to learn with unpaired video and text data using an adversarial formulation. We experimentally confirmed that learning such an embedding space benefits three

Table 7: Comparison of several models for unseen activity captioning using the ActivityNet dataset, using METEOR and CIDEr scores. This evaluation was done on 10 unseen classes held out during training. Higher values are better.

|  | METEOR | CIDEr |
|---|---|---|
| Fixed Text Representation | 3.64 | 8.95 |
| Joint | 4.21 | 9.23 |
| All (paired) | 5.31 | 11.21 |
| All (paired + unpaired) | 6.89 | 13.95 |

difficult tasks (i) zero-shot activity classification, (ii) unsupervised activity discovery, and (iii) unseen activity captioning. We find that the use of related unpaired data is beneficial. We presented several strategies for obtaining unpaired data and confirmed the benefit of adding additional, relevant unpaired data.

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

Table 8: Comparison of temporal pooling methods for 5 unseen classes in the ActivityNet dataset.

|                          | Accuracy |
|--------------------------|----------|
| Max Pooling              | 23.4     |
| Sum Pooling              | 24.1     |
| LSTM                     | 42.3     |
| Temporal Attention Filters | 55.2   |

Table 9: Comparison of different ratios of paired and unpaired data methods for 5 unseen classes in the ActivityNet dataset.

| Paired/Unpaired | Accuracy |
|-----------------|----------|
| 100% / 0%       | 74.2     |
| 75% / 25%       | 73.2     |
| 50% / 50%       | 69.7     |
| 25% / 75%       | 62.6     |
| 0% / 100%       | 24.5     |

## A  ADDITIONAL EXPERIMENTS

### A.1  COMPARISON OF TEMPORAL POOLING METHODS

To confirm that temporal attention is beneficial, we compare different forms of temporal pooling (i) max-pooling, (ii) sum-pooling, (iii) LSTM, and (iv) temporal attention filters (Piergiovanni et al., 2017). In Table 8, we compare these temporal pooling methods learning the joint embedding space. We confirm that using the temporal attention filters performs best.

### A.2  COMPARISON OF DIFFERENT RATIOS OF PAIRED AND UNPAIRED DATA

We compare different ratios of paired and unpaired data to see how much paired data we require and how much unpaired data is beneficial. For these experiments, we use all the loss terms (i.e., what provided us the best results). Note that in these experiments, the total number of samples was the same for each method (40k examples) so that we can directly compare the effects of unpaired data vs. paired data. Thus not all the available data was used.

In Table 9, we show the results. We find that using no paired data results in nearly random performance, but using using some paired data greatly improves the embedding space. The model using 100% paired data performs best, as all the others are using less overall paired data.

We also compare augmenting our 40k paired training samples with different amounts of unpaired data. Since UCF101 and HMDB only have 13k and 7k examples, to get up to 60k samples, we also use videos from the Kinetics dataset (Carreira & Zisserman, 2017). The results, shown in Table 10, show that adding the initial 10k samples is most beneficial, while additional samples do not seem to meaningfully improve results. However, due to our training method where each batch consists of 50% paired data and 50% unpaired data, the additional unpaired data does not harm results either.

Table 10: Comparison using 40k paired examples and varying amounts of unpaired samples for 5 unseen classes in the ActivityNet dataset.

| Unpaired Samples | Accuracy |
|------------------|----------|
| 0                | 77.1     |
| 10k              | 82.4     |
| 20k              | 83.9     |
| 40k              | 83.6     |
| 60k              | 83.5     |

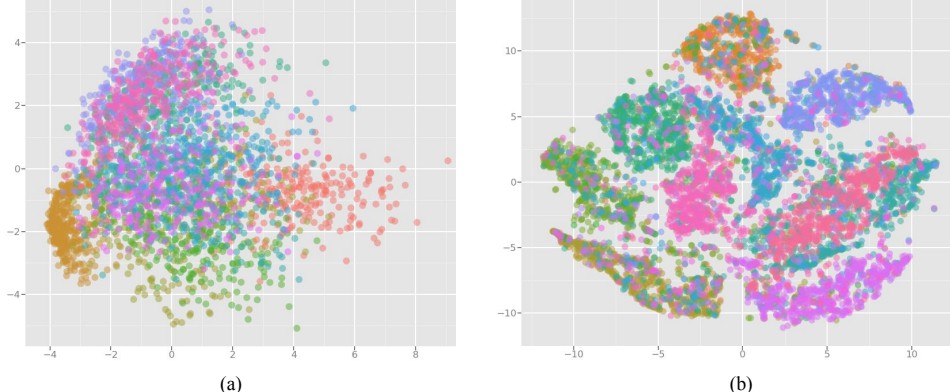

(a)          (b)

Figure 6: t-SNE mapping of **(a)** fixed text representation and **(b)** joint embedding with all paired losses for the MLB-YouTube dataset. The joint embedding space provides most distinct representations for the activities. Each color represents the activity class of the video (e.g., swing, hit, foul ball, etc.).

Table 11: Comparison of several models for standard, seen video captioning using the MLB-YouTube dataset, using Bleu, METEOR and CIDEr scores. Higher values are better.

|  | Bleu | METEOR | CIDEr |
|---|---|---|---|
| Fixed Text Representation | 0.12 | 0.04 | 0.12 |
| Joint Representation | 0.14 | 0.08 | 0.15 |
| Joint + all paired | 0.15 | 0.10 | 0.18 |
| Joint + paired + unpaired | 0.10 | 0.02 | 0.08 |

### A.3 MLB-YOUTUBE CAPTIONS

In Fig. 6, we compare t-SNE embeddings of the fixed text representation and our joint embedding space. This visually shows that learning a joint embedding space gives more distinct class distributions.

#### A.3.1 MLB-YOUTUBE CAPTIONS

As a baseline for the MLB-YouTube captions dataset, we compared several different models for standard video captioning (i.e., all activity classes are seen). This task is quite challenging compared to other datasets as the announcers commentary is not always a direct description of the current events. Often the announcers tell loosely related stories and attempt to describe events differently each time to avoid repetition. Additionally, the descriptions contain on average 150 words for each 30 second interval and current captioning approaches usually only trained and tested on 10-20 word sentences. Due to these factors, this task is quite challenging the standard evaluation metrics do not account for these factors. In Table 11, we report our results on this task.

## B HMDB AND UCF101 SENTENCES

For the HMBD and UCF101 datasets, we created sentences to describe each activity class. Our sentences descriptions are included in this appendix.

These sentences are written for each activity class (by randomly selecting a single video per class) and are shared for all instances of the activity. Depending on what video was randomly chosen for the class, some sentences describe the actor as a 'man', 'woman', or 'person' which could confuse the model. Ideally, the CNN embedding needs to learn to ignore the impact of such pronoun changes.

We conducted experiments comparing randomly replacing the pronouns to determine if there was any bias introduced by the pronouns. We show the results in Table 12. We find that the choice of pronouns does not impact performance, as our model automatically learns to focus more on verbs rather than pronouns. When examining the temporal attention filters on the sentences, we found that

Table 12: Comparison of various pronouns on the UCF101 dataset with 50 unseen classes.

|                   | Accuracy |
|-------------------|----------|
| Baseline Sentences | 33.4 |
| All 'man' | 33.2 |
| All 'woman' | 33.3 |
| All 'person' | 33.4 |
| Random pronoun | 33.4 |

they placed very little 'attention' on the start of the sentence, where the pronoun usually is, suggesting that the pronoun has very little effect on the embedding space we learned.

HMDB:

1. chew: a woman is chewing on bread
2. golf: a man swings a golf club
3. sword exercise: a person is playing with a sword
4. walk: a person is walking
5. jump: a person jumps into the water
6. pour: a man pours from a bottle
7. laugh: a man is laughing
8. shoot gun: a person rapidly fires a gun
9. run: a person is running
10. turn: a person turns around
11. ride bike: a man is riding a bike on the street
12. swing baseball: a boy hits a baseball
13. draw sword: a person draws a sword
14. sit: a person sits in a char
15. fencing: two men are fencing
16. dribble: a boy dribbles a basketball
17. stand: a person stands up
18. pushup: a man does pushups
19. sword: two people are fighting with swords
20. pullup: a boy does pullups in a doorway
21. smile: a man smiles
22. shake hands: two people shake hands
23. shoot ball: a person shoots a basketball
24. kick: a person kicks another person
25. somersault: a person does a somersault
26. flic flac: a boy does a backflip
27. hug: two people hug
28. hit: a boy swings a baseball bat
29. dive: a person jumps into a lake
30. drink: a man drinks from a bottle
31. punch: a woman punches a man
32. wave: a person waves their hand
33. talk: a person is talking

34. kiss: a man and woman kiss
35. catch: a boy catches a ball
36. smoking: a woman smokes a cigarette
37. eat: a man eats pizza
38. throw: a person throws a ball
39. climb stairs: a man is running down the stairs
40. kick ball: a person kicks a soccer ball
41. ride horse: a girl is riding a horse
42. fall floor: a man is pushed onto the ground
43. brush hair: a girl is brushing her hair
44. situp: a man does situps
45. cartwheel: a guy runs and jumps and flips
46. pick: a man picks a book
47. push: a boy pushes a table
48. climb: a man is climbing up a wall
49. handstand: three girls do handstands
50. clap: a woman claps her hands
51. shoot bow: a person shows a bow and arrow

UCF101:

1. MilitaryParade: people are marching and waving a flag
2. TrampolineJumping: kids are jumping on a trampoline
3. PlayingDaf: a person moves a circle and hits it
4. SalsaSpin: poeple are dancing and spinning
5. CuttingInKitchen: a person is in the kitchen using a knife
6. ApplyEyeMakeup: a woman is putting on makeup
7. PlayingViolin: a person plays the violin
8. YoYo: a person plays with a yoyo
9. PlayingCello: a person is playing the cello
10. Bowling: a person is bowling
11. UnevenBars: a woman is spinning and flying on bars
12. BalanceBeam: a woman is on the balance beam
13. SkyDiving: people are falling out of the sky
14. SumoWrestling: two fat people are wrestling
15. PushUps: a man does pushups
16. FloorGymnastics: a girl does gymnastics
17. ApplyLipstick: a woman is putting on lipstick
18. BreastStroke: a woman is swimming
19. GolfSwing: a man swings a golf club
20. PlayingDohl: a person hits on a drum
21. HorseRiding: a woman rides a horse
22. PlayingFlute: a person blow into a flute
23. PizzaTossing: a man is making a pizza
24. CleanAndJerk: a person is lifting weights

25. WritingOnBoard: a person is writing on the wall

26. CricketShot: a person hits a ball with a bat

27. FieldHockeyPenalty: a girl in the field shoots a ball

28. HammerThrow: a person spins and throws an object

29. BodyWeightSquats: a man is squatting

30. CliffDiving: a person jumps off a cliff

31. Typing: a person is typing at a computer

32. MoppingFloor: a man mops the floor

33. TaiChi: people are doing tai chi

34. PlayingPiano: a person plays piano

35. Punch: someone punches another person

36. Nunchucks: a person swings nun chucks

37. RopeClimbing: a person climbs a rope

38. Swing: a baby is swinging

39. Knitting: a woman is knitting

40. Rafting: people are rafting on a river

41. PlayingGuitar: a person strums a guitar

42. ShavingBeard: a man shaves his beard

43. JugglingBalls: a person is juggling balls

44. Diving: a boy dives into a pool

45. JumpingJack: a person jumps and swings his arms

46. VolleyBallSpiking: people hit a volleyball

47. PoleValut: a person runs with a pole and launches into the air

48. SkateBoarding: a man is skateboarding

49. BoxingPunchingBag: a man is punching a bag

50. IceDancing: people are ice skating

51. WallPushups: a person does pushups against a wall

52. FrisbeeCatch: a person jumps and catches a frisbee

53. Drumming: people are drumming

54. JumpRope: a girl is jumping rope

55. HeadMassage: a person gets their head massaged

56. PlayingTabla: a person plays two drums

57. TableTennisShot: people are playing table tennis

58. PommelHorse: a person spins around on their hands

59. HighJump: a man jumps over a bar and lands on his back

60. BasketballDunk: a man jumps and dunks the basketball

61. BoxingSpeedBag: a man punches a bad in the air quickly

62. PullUps: a person does hangs on a bar and pulls up

63. RockClimbingIndoor: a person is climbing up rocks

64. BlowingCandles: a boy blows out candles on a cake

65. Skiing: people are skiing on a mountain

66. WalkingWithDog: a person walks a dog

67. Basketball: men are playing basketball

68. SoccerJuggling: a person is playing with a soccer ball

69. Fencing: people are fencing

70. Billiards: a man is playing billiards

71. BaseballPitch: a man throws a baseball

72. BlowDryHair: a woman is drying her hair

73. CricketBowling: a person throws a cricket ball

74. BandMarching: people are walking down the street playing music

75. PlayingSitar: a person plays a funny guitar

76. ThrowDiscus: a person spins and throws a disk

77. StillRings: a man holds in the air on rings

78. Lunges: a person bends to the ground with one knee

79. Skijet: a person rides a jetski in the ocean

80. BabyCrawling: a baby is crawling on the floor

81. Mixing: a woman is mixing in a bowl

82. Hammering: a person is hitting nails with a hammer

83. Shotput: a person spins and launches a ball

84. Archery: a man shoots a bow and arrow

85. Surfing: a man is surfing in the ocean

86. FrontCrawl: a person is swimming freestyle

87. HulaHoop: a person spins a hoop around their waist

88. JavelinThrow: a person throws a spear

89. Rowing: people are in a canoe and rowing

90. Kayaking: a person is kayaking on a lake

91. ParallelBars: a man does gymnastics on the parallel bars

92. HorseRace: horses are racing around a track

93. HandstandWalking: a person stands on their hands and walk

94. BrushingTeeth: a boy brushes his teeth

95. LongJump: a person runs and jumps into a sand pit

96. Biking: people are riding bikes

97. HandstandPushups: a person does pushups upside down

98. BenchPress: a man is lifting weights

99. Haircut: a person is getting a hair cut

100. TennisSwing: a woman hits a tennis ball

