# OpenReview forum: "Unseen Action Recognition with Unpaired Adversarial Multimodal Learning"
_ICLR.cc/2019/Conference_

### Official Review · AnonReviewer3 · 2018-10-30
**limited novelty and wrong experimental settings**

**Rating:** 4
**Confidence:** 5

**Review:**

This paper proposes a joint embedding model that aligns video sequences with sentences describing the context (caption) in a shared embedding space. With the space, various tasks such as zero-shot activity recognition and unseen video captioning can be performed. The problem tackled in this paper is interesting. However, the approach proposed is limited in novelty and there are some serious flaws in the experimental settings. So overall, this paper is not yet ready for publication.

Pros:

•	The overall bidirectional encoder-decoder architecture for learning a shared embedding space is sensible. It is also interesting that adversarial training is introduced so that unlabelled data can be utilized.
•	Additional annotations are provided to two activity recognition datasets, creating new benchmarks.
Cons
•	My first concern is the limited novelty of the work. Although I am not aware of a joint embedding learning model that has exactly the same architecture and formulation, the model is closely related to many existing ones both in zero-shot learning and beyond. More specifically,
o	The overall framework is similar to “correlational neural networks”, Neural Computation, 2016 by Chandar et al. This should be acknowledged.
o	The connections to CyclyGan and its variants for image-to-image style transfer is obvious, as pointed out by the authors.
o	More importantly, there are quite a few closely related zero-shot learning (ZSL) papers published recently. Although they focus on static images and class name, rather that image sequences and sentences, I don’t see any reason why these models cannot be applied to solve the same problem tackled in this paper. In particular, the autoencoder architecture was first used in ZSL in E. Kodirov, T. Xiang and S. Gong, "Semantic Autoencoder for Zero-Shot Learning", in IEEE Conference on Computer Vision and Pattern Recognition (CVPR), Honolulu, Hawaii, July 2017. This work is further extended in Chen et al, “Zero-Shot Visual Recognition using Semantics-Preserving Adversarial Embedding Network”, cvpr18, now with adversarial learning. Similarly, variational autoencoder is adopted in Wang et al, Zero-Shot Learning via Class-Conditioned Deep Generative Models, AAAI 2018.  Note that the joint embedding spaces in these studies are the semantic spaces – attribute or word vector spaces representing the classes. In contrast, since the semantic modality is a variable-length word sequences, this is not possible, so a third space (other than the visual feature space or semantic space) is used as the embedding space. Beyond these autoencoder based models, there are also a number of recent ZSL works that use a conditional generative model with adversarial loss. Instead of learning a joint embedding space where the visual and text modalities are aligned and compared for recognition, these works use the text modality as condition to the generative model to synthesize visual features for the unseen classes followed by a conventional supervised classifier. The representative one of this line of work is Xian et al, “Feature Generating Networks for Zero-Shot Learning, cvpr18”.
o	In summary, there are too many existing works that are similar to the proposed one in one or more aspects. The authors failed to acknowledge most of them; moreover, it is not argued theoretically or demonstrated empirically, why combining different approaches together is necessary/making fundamental differences.

•	My second main concern is the experiment setting. This paper adopts a conventional ZSL setting in two aspects: (1) the visual features are obtained by a video CNN, I3D, which is pretrained on the large (400 or 600 classes depending on which version is used) Kinetics dataset. This dataset have classes overlapping with those in ActivityNet, HMDB and UCF101. So if these overlapped classes are used in the unseen class partition, then the ZSL assumption (the target classes are ‘unseen’) is violated. (2): The test data only contains unseen class samples. In practice, one will face a test set composed of a mix of seen and unseen classes. Under this more realistic setting (termed generalized ZSL in the ZSL community), a ZSL must avoid the bias towards the seen classes which provide the only visual data available during training. The two problems have been identified in the ZSl community when static images are considered. As a result, the conventional setting has been largely abandoned in the last two years and the ‘pure’ and ‘generalized’ settings become the norm; that is, there is no overlapping classes between the test classes and classes used to pretrain the visual feature extraction network; and both seen and unseen class samples are used for testing. The ZSL evaluation is only meaningful under this more rigorous and realistic setting. In summary, the experimental results presented in this paper are obtained under the wrong setting and the proposed model is not compared with a number of closely related ZSL models listed above, so it is not possible to judge how effective the proposed model is.

---

> ### Author Response · Authors · 2018-11-27
> **Addressing the experimental setting, our contribution and previous works**
>
> We thank the reviewer for the comments. We would like to address an important misunderstanding regarding the contribution of the paper. We also revised the paper to include the results with the new experimental setting suggested by the reviewer.
>
> - The contribution of the paper and comparison to the previous work:
>
> We want to clarify that our primary, novel contribution is the addition of the *unpaired* data to learn a shared embedding space. We added missing references, but we would like to also mention that none of the previous work used unpaired data to learn the shared embedding or to do zero-shot action recognition.  We believe this is the first paper which attempted to use unpaired data in addition to conventional paired data, and this is what makes our approach different from the previous ones. The proposed model is not just a combination of previous methods, but is what we fundamentally designed to allow the learning from the unpaired examples. The difference to the previous models is whether we can make the embedding space benefit from unpaired data or not and we confirming such capability experimentally.
>
> In the revised version of the paper, we have added the references to the missing works, and do experimental comparisons to those approaches (Tables  and 2). We also compare ours to the use of different aspects of the adversarial loss on paired data (Table 1). Again, the major difference from all existing zero-shot works is that we use unpaired data and we experimentally find it benefits learning the embedding space. We also have further added a section exploring the use of different unpaired data (Section 4.3).
>
> - Experimental setting without overlapping pre-training.
>
> Following the suggestion of the reviewer, we revised *all* our experimental results by using a version of I3D trained on Kinetics with the classes in ActivityNet, HMDB51 and UCF101 withheld. That is, we trained I3D visual features while only using videos of seen classes. We used the Kinetics-600 dataset and after removing all the classes with overlapping classes with ActivityNet, HMDB51 and UCF101, this resulted in 478 activity classes and 278k training videos. We re-ran all the experiments using features from this new version of I3D and updated the results in the paper. We note that this did not significantly impact results.
>
> - Experimental setting to test our approach with both unseen and seen activity classes
>
> Yes, the previous zero-shot works on video only tested on the unseen/withheld classes (i.e., Sec. 3 of Xu et al. “Transductive Zero-Shot Action Recognition by Word-Vector Embedding”). We chose to follow this setting in order to directly compare ours to the previous works. Following the suggestion of the reviewer, in the revised paper, we have added tables (Tables 3 and 4) evaluating our approach on the entire datasets (both unseen and seen classes) as well. Please note that these results are not directly comparable to results reported in the previous papers on zero-shot activity recognition which followed the old setting.

---

### Official Review · AnonReviewer1 · 2018-11-02
**Contribution of temporal attention is not evaluated on all tasks - decisions in experiments not justified throughout and take-home message missing**

**Rating:** 5
**Confidence:** 4

**Review:**

The paper attempts multimodal representation of video and text through an attention layer that allows weighted temporal pooling. The approach was tested on a collection of datasets including a newly introduced dataset, with the embedding and evaluated on three tasks: zero-shot classification, activity clustering and captioning.

The paper is easy to read in general and the approach is scientifically sound. The need for an autoencoder in multimodal embedding has been proven for a variety of modalities including image-text, video-text, image-audio and video-audio. The contribution here is thus focused on temporal pooling through a learnt attention layer.

However, the paper has a mix of tasks (3 tasks tested), without a conclusive understanding of the effect of the various loss functions on the learnt space. As the importance of various losses changes per task and dataset, the take-away message from the work is not obvious. Additionally, using unpaired data, proposed through a large-scale dataset is not obvious. The paper concludes that related data is required but how related data can be collected remains unexplored.

The evaluation for the unsupervised discovery seems biased – 1NearestNeighbour is used as opposed to the more balanced mAP on ranking all test sequences as opposed to top-1.

The collected dataset, which is a contribution of the paper is also poorly explained. The authors collect ‘dense annotations’ but it is not clear how many annotators were used, and what instructions they were given. The paper does not give examples of the collected annotations and how these differ from previous annotations available with the dataset (Fig 4).

Appendix 1 concludes with sentences proposed to annotate UCF. These seem to apply per action and it’s not clear how they scale to the different instances, e.g. Action Surfing (85) is assigned to a male caption ‘a man is’, action 100 to a woman and action 96 to groups of people ‘people are riding’. This distinction is not obvious in all the instances of the dataset and such captioning might have significantly biased the results.

Overall, there is little explanation of the decisions made to produce the comparative results. The novelty is limited to the attention pooling, which is not evaluated on all the three tasks.

---

> ### Author Response · Authors · 2018-11-27
> **Clarifying our contribution and addressing your concerns**
>
> Thank you for the review, we revised our paper with additional experiments to address your concerns. Please find our detailed answers to your comments below.
>
> - The main contribution of the paper:
>
> We first want to clarify that the use of the attention pooling is not the main novel contribution of the paper. Our primary contribution is the use of unpaired, multimodal data for learning an embedding space that generalizes to unseen actions. We compare the use of the unpaired data on all tasks and have added a new section comparing different strategies for obtaining unpaired data. To our knowledge, this is the first paper to explicitly use unpaired examples for multimodal embedding learning for zero-shot action classification. We changed title of our paper to emphasize this more explicitly.
>
> - Conclusive findings:
>
> As we illustrate in Tables 1-7, our conclusion is that (1) learning of the shared multimodal embedding becomes possible with our approach, that (2) it allows the embedding to benefit from unpaired data (unlike previous works), and that (3) using all loss functions we introduced jointly benefits the unseen action recognition the most. We revised the paper to make this more explicit. As we show in Table 1, we report finding on the effect of the loss functions, and we find that the use of unpaired data is beneficial and all the terms are beneficial. Also note that we added an experimental results testing the use of different amount of unpaired training videos (Section A.2).
>
> - How do we obtain related vs. unrelated data ?
>
> Basically, related data means that they are the videos from the same context (e.g., baseball), while unrelated data means that they are from very different context (e.g., baseball in MLB-YouTube vs. office activities in Charades) not sharing any similar video segments or text sentences. In the revised paper, we have added a section and experiments comparing different sources/strategies for obtained unpaired data (Table 4). We were able to observe that the performance drops when adding unpaired random sentences or random dictionary definitions. On the other hand, when we only add sentences of 'verb' definitions which are relevant to actions by their nature, there was a gain in performance. Unpaired videos usually helped since they at least benefit learning better feature representations.
>
>
> - New dataset annotation details:
>
> The MLB-YouTube Captions dataset we collected contains annotations that are transcriptions of the *real MLB broadcast* announcers commentary. No additional annotator was used. Figure 4 contains examples of the annotations. Previous MLB-Youtube dataset annotations were simply class labels and time intervals (i.e., no text or language data). We made this more explicitly clear in the revised paper.
>
> - "The evaluation for the unsupervised discovery seems biased".
>
> Following the suggestion of the reviewer, we added mAP results to the unsupervised activity discovery methods (Table 5). It confirms similar findings.
>
> - Biased UCF sentences?
>
> Regarding the activity sentences in the appendix, they are per-action. The captions were obtained by an annotator writing a sentence for a random video from the dataset. We conducted additional experiments, and found that the use of “man,” “woman,” “boy,” “girl,” “person,” or “people” did not meaningfully impact results. Specifically, we conducted experiments by randomly replacing those pronouns in the sentences. We added the results in Table 12 in the appendix.
>
> We further examined the embedding of the sentences by switching the pronouns, and found that they all were quite close together (average distance of 0.08 for the same sentence with different pronouns, while the average distance of different activity sentences was 0.86 in the embedding space). We are focusing on the "action recognition" problem where the verbs are the most meaningful information, and we confirm that our CNN embedding learns to ignore the impact of pronoun changes.

---

### Official Review · AnonReviewer2 · 2018-11-02
**Shared video-text embedding learnt with AEs and Adversarial learning applied to three diverse tasks. The paper has novel aspects and is well evaluated.**

**Rating:** 7
**Confidence:** 4

**Review:**

Summary:
The paper aims to learn a common embedding space for video appearance and text caption features. The learned shared embedding space then allows multiple applications of zero-shot activity classification, unsupervised activity discovery and unseen activity captioning.

The method is based on two autoencoders which have a common intermediate space. The losses optimized encourage the standard unimodal reconstructions in the AEs, along with joint embedding distances (appearance and text of the same video are encoded close by) as well as cross domain mapping (video encoding generates text and vice-versa), and cycle consistency. Apart from these additional supervised losses, unsupervised losses are added with adversarial learning which aim to bring the video and text encoding distributions in the common space close, as well as the standard real and generated video/text distributions close by adding corresponding discriminators (like in GANs). The whole system is learned end-to-end in two phases, first with supervised paired data and then with all paired and unpaired data.

The experiments are shown on four datasets: ActivityNet, HMDB, UCF101, MLB-YouTube

Positives:
- The problem of multimodal learning is an interesting and challenging problem
- The paper is novel; while the idea of a joint shared embedding space is not new this paper adds new losses as summarized above and shows reasonably convincing empirical results
- The results are shown for diverse applications which highlight the generality of the method
- The use of unpaired/unsupervised data is also relatively less explored which this paper incorporates as well
- The empirical results given are convincing, eg Tab1 gives a good ablation study showing how the different components affect the performance. SoA comparison are given on a standard task (however see below)

Negatives:
- Comparison with state of the art result Tab2 should also contain the features used. The performances might just be higher due to the better features used (Kinetics pretrained I3D). Please give a self implemented baseline method with same features but some standard loss in the shared space to give an idea of the strength of the features.
- Ideally features provided by previous papers’ authors should be used if available and it should be shown that the proposed method improves results.

Overall the paper is well written and had novel aspects which are convincingly evaluated on challenging and diverse tasks.

---

> ### Author Response · Authors · 2018-11-27
> **Thanks for the review, we address your concerns in the comment**
>
> We thank the reviewer for the review. We revised our paper to address a couple of concerns mentioned. Please also find our answers below.
>
> 1. We have added the feature type used to Table 2. Most previous works all used Improved Dense Trajectories (IDT) as the feature. Following the suggestion of the reviewer, we also implemented and added the results of our method using IDT features (with both paired and paired+unpaired data). We were able to observe that, while using the same IDT features, our method outperforms the previous works.
>
> 2. Our baseline using fixed text and that using joint losses are very similar to previous standard loss functions. Table 1 shows the superiority of our method. We have further added the results in Table 1 using the "triplet loss", which is another standard loss for embedding spaces. Note that the triplet loss requires positive+negative triplets, while our approach uses less annotated data (only matching text+video and unpaired text and video).
>
> Please let us know if this addresses your concerns and let us know if you have any others.

---

### Meta-Review · Area_Chair1 · 2018-12-11
**decision**

**Confidence:** 4
**Recommendation:** Reject

**Metareview:**

The paper received mixed reviews. The proposed ideas are reasonable and it shows that unpaired data can improve the performance of unseen video (action) classification tasks and other related tasks. The authors rightfully argue that the main contribution is the use of unpaired, multimodal data for learning a joint embedding (that generalizes to unseen actions) with positive results, but not the use of attentional pooling mechanism. Despite this, as the Reviewer3 points out, technical novelty seems minor as there are quite many papers on learning joint embedding for multimodal data. Many of these works were evaluated for fine-grained image classification setting, but there is no reason that such methods cannot be used here. The revision only compares against methods published in 2017 or before. So more comprehensive evaluation would be needed to fully justify the proposed method. In addition, it seems that the proposed method has fairly marginal gain for the generalized zero-shot learning setting. Overall, the paper can be viewed as an application paper on unseen action recognition tasks but the technical novelty and more rigorous comparisons against recent related work are somewhat lacking. I recommend rejection due to several concerns raised here and by the reviewers.